# The Impact of a Pharmacist-Led Intravenous to Oral Switch of Metronidazole: A Before-and-After Study

**DOI:** 10.3390/antibiotics11101303

**Published:** 2022-09-25

**Authors:** Mahdi Algargoosh, Stephen Ritchie, Eamon Duffy, Bert Van der Werf, Mark Thomas, Nataly Martini

**Affiliations:** 1Department of Pharmacy, Auckland City Hospital, Auckland 1023, New Zealand; 2School of Pharmacy, Faculty of Medical and Health Sciences, University of Auckland, Auckland 1023, New Zealand; 3Department of Molecular Medicine and Pathology, Faculty of Medical and Health Sciences, University of Auckland, Auckland 1023, New Zealand; 4Department of Epidemiology and Biostatistics, Faculty of Medical and Health Sciences, University of Auckland, Auckland 1023, New Zealand

**Keywords:** antimicrobial stewardship, intravenous to oral switch, pharmacist-led, switch therapy

## Abstract

(1) Background. Intravenous (IV) to oral switch (IVOS) of antibiotics can reduce the length of hospitalisation, risk of IV catheter complications, and hospital costs. Pharmacists can play an instrumental role in implementing an IVOS initiative. The aim of this study is to evaluate the impact of pharmacist-led IVOS of metronidazole. (2) Method. This was an observational study conducted in a New Zealand hospital. During a 3-month intervention period, pharmacists identified patients receiving IV metronidazole; then initiated an IVOS for patients who met the criteria. The comparator groups were patients who were not switched by pharmacists in the post-intervention (post-IVOS) group, or patients treated with either IV or oral metronidazole prior to the intervention (pre-IVOS). Primary outcome measures were switch rate and duration of IV metronidazole treatment. Secondary outcome measures were readmission and/or repeat surgery within 90 days of discharge and the length of hospital stay. (3) Results. In total, 203 patients were included: 100 in the pre-IVOS and 103 in the post-IVOS groups. Pharmacists switched 63/93 (67.7%) of eligible patients to oral metronidazole in the post-IVOS period. Only 9/89 (10.1%) of IVOS eligible patients were switched in the pre-IVOS group. In the post-IVOS group, the mean duration of IV metronidazole treatment in patients switched by pharmacists was shorter than in those who were not switched by pharmacists (2.5 ± 2.8 days vs. 4.8 ± 5.9 days, *p* = 0.012). No significant difference was found in readmission or repeat surgery within 90 days of discharge for patients switched by pharmacists versus patients who were not switched by pharmacists. (4) Conclusion. Our data have demonstrated successful implementation of the hospital-approved pharmacist-led IVOS service.

## 1. Introduction

Antimicrobial stewardship (AMS) programmes play a vital role in reducing inappropriate antimicrobial use by providing guidance to optimise the selection, dose, route, and duration of antimicrobial treatment. Antimicrobial stewardship programmes minimise antibiotic-related harm and prevent prolonged lengths of stay secondary to adverse drug effects [1,2]. Intravenous (IV) antibiotics are generally more expensive than oral antibiotics, and the use of IV lines can lead to significant complications such as catheter-associated infections and thrombophlebitis [3,4]. When indicated, switching to oral therapy minimises the risk of those complications, reduces antibiotic preparation time, eases administration, improves patient comfort, and decreases length of hospital stay [5,6,7,8].

Traditional IV to oral switch (IVOS) programmes have usually been reliant on the need for doctors’ approval to initiate the switch [9,10]; this was shown to be time-consuming for both pharmacists and doctors, and did not guarantee an IVOS [7,11]. As a result, the switch is often delayed for patients who already meet the switch criteria [9,10,12,13,14]. A study in 2016 showed that two-thirds of patients continued IV antibiotics for 72 h after becoming eligible for a switch to an oral formulation [9]. Other barriers to implementing timely IVOS of antibiotics include misconceptions that oral antibiotics are less effective, and a lack of awareness of IVOS guidelines [15]. 

A more efficient strategy is to allow pharmacists to initiate the switch independently [5,7,14]. Clinical pharmacists undergo extensive training and certification to ensure clinical competency and up-to-date clinical knowledge [16]; therefore, they are well placed to make these changes. The duties of clinical pharmacists are central to AMS, which include optimisation of dosing through knowledge of the pharmacokinetics, pharmacodynamics and economics of medicines [16,17,18]. Pharmacist-led IVOS programmes have resulted in cost-minimisation and shorter hospital stays in patients converted to oral therapy [5,7,16,19]. Studies that evaluated pharmacist-led IVOS of fluoroquinolones reported shorter durations of IV therapy, and no clinically significant differences in clinical efficacy [12,20]. 

In May 2019, a pharmacist-led IVOS programme was introduced at Auckland City Hospital, New Zealand, with metronidazole nominated as the initial antibiotic of focus. Metronidazole is an antibacterial and antiprotozoal drug commonly used for the prophylaxis and treatment of anaerobic bacterial infections including intra-abdominal infections, deep neck space infections, pelvic inflammatory disease, lung abscess, and amoebiasis [21,22]. Metronidazole was chosen in this study due to its high bioavailability (>90%) and tissue penetration to infection sites [23], and the ability to tolerate oral food or medicines is the sole eligibility criteria for switching IV metronidazole to oral for non-septic patients [23]. To date, most published studies of pharmacist-led IVOS have used fluoroquinolones [7,12,19,24,25]; only a few have used other antimicrobials such as metronidazole [5,26,27].

The objective of this study was therefore to measure the pharmacist-led switch rate, and the duration of IV metronidazole treatment in response to the IVOS metronidazole programme at ACH. We hypothesised that pharmacist-led IVOS of metronidazole would increase the switch rate, and reduce the duration of IV metronidazole treatment without compromising patients’ clinical outcomes.

## 2. Materials and Methods

Auckland City Hospital is an adult secondary, tertiary and quaternary referral centre hospital, serving a local population of approximately 540,000 in Auckland, New Zealand.

The metronidazole IVOS programme was implemented by the AMS committee using the following simple criteria: tolerating oral medicines or food in non-ICU wards and requiring pharmacists to document the prescription change in the patient’s clinical record and inform the patient and responsible medical team. All clinical pharmacists were trained to implement the IVOS programme prior to its introduction, and their concerns were addressed through discussions, in-service training, and ad hoc support from AMS committee members. The IVOS programme was supported by the Chief Medical Officer, the AMS Committee and the clinical services with the highest use of metronidazole (General Surgical, Gynaecology, and Gastroenterology). Ethics approval to study the impact of the IVOS was provided by The Auckland Health Research Ethics Committee (Reference: 00082).

### 2.1. Study Participants and Design

A prospective review of adult patients (18 years and older) on General Surgical, Gynaecology, and Gastroenterology wards, prescribed either IV or oral metronidazole, was undertaken over three months (10 June to 30 September 2019). No specific exclusions were applied. Six clinical pharmacists identified patients receiving metronidazole during their routine clinical activities and recorded clinical information (patient demographic details, diagnosis, oral intake status, and duration and number of doses of IV and oral metronidazole treatments) using Qualtrics^XM^ (Qualtrics, Provo, UT, USA). Information regarding length of hospital stay, readmission, repeat surgery, percutaneous drainage procedures and all-cause mortality up to 90 days were generated from the hospital administration systems.

Patient data from the post-intervention (post-IVOS) study, described above, were compared with those from age, sex, ethnicity, and diagnosis matched controls obtained prior to initiation of the IVOS programme (1 January to 31 December 2018). The pre-intervention (pre-IVOS) cohort was identified by hospital data analysts and clinical data was collected. Pre-IVOS patients were reviewed sequentially, and the first 100 patients who received metronidazole therapy were selected for inclusion. IVOS eligibility was determined based on whether patients were receiving other oral medicines whilst on IV metronidazole.

### 2.2. Outcome Measures

The primary outcome measures were pharmacist-led switch rate, and the duration of IV metronidazole treatment. The secondary outcome measures were readmission and/or repeat surgery within 90 days of discharge, and the length of hospital stay.

### 2.3. Analysis

In the absence of local data to enable a formal sample size calculation, an a priori inclusion target of 200 eligible patients (100 patients in each group) was used, based on reports from similar studies that investigated the effects of a similar intervention [5,14,19]. 

Patients in the pre- and post-IVOS groups were stratified into four metronidazole treatment subgroups for data analysis: IV only; IVOS; IVOS initially, then reversion to IV; and oral only.

Comparisons were made between patients who were switched by a pharmacist and patients who were not switched by a pharmacist in the post-IVOS group, and between the pre- and post-IVOS groups.

Kaplan–Meier estimates were used to analyse the effect of study phase and pharmacist-led switch on time to repeat surgery and readmission within 90 days post-discharge. Kaplan–Meier estimates and plots were constructed in R language and environment for statistical computing and graphics version 4.03 (Vienna, Austria) [28] using the package survminer [29].

All values were reported as the mean ± standard deviation (SD) unless otherwise specified. A *p*-value <0.05 was considered statistically significant for all analyses. 

## 3. Results

Two hundred and three patients treated with either IV or oral metronidazole were included in the study: 100 patients in the pre-IVOS group, and 103 in the post-IVOS group. Characteristics of the patients are shown in Table 1.

In the pre-IVOS group, only 9/89 (10.1%) who met the switch criteria were switched to oral metronidazole by doctors. In the post-IVOS period, 93/103 (90.3%) were eligible for IVOS; of theses, pharmacists switched 63/93 (67.7%), and doctors switched 2/93 (2.2%). 

The uptake of the IVOS by pharmacists was slow; only 3/18 (16.7%) of eligible patients were switched during the first month of the intervention period. This increased over the intervention period with 14/18 (77.8%) of eligible patients switched in the final month. 

No significant difference in the mean duration and number of IV metronidazole doses between the pre- and post-IVOS (3.6 ± 2.8 days vs. 3.4 ± 4.4 days, *p* = 0.34; 8.7 ± 6.8 doses vs. 7.8 ± 8.5 doses, *p* = 0.19, respectively). In the post-IVOS group, the mean duration and number of IV metronidazole doses in patients switched by pharmacists were considerably lower than in those who were not switched by a pharmacist (2.5 ± 2.8 days vs. 4.8 ± 5.9 days, *p* = 0.012; 6.2 ± 6.9 doses vs. 10.3 ± 10.1 doses, *p* = 0.014, respectively).

In the post-IVOS group there was no difference in the 90-day readmission rates for patients switched to oral metronidazole by pharmacists 15/63 (23.8%) versus those not switched by pharmacists 9/40 (22.5%; *p* = 0.6). Similarly, the rates of repeat surgery did not differ between patients who were switched by pharmacists 5/63 (7.9%) and patients who were not switched 4/40 (10.0%, *p* = 0.56). There was no difference in the 90 days readmission rate for patients in the pre-IVOS group versus post-IVOS group (18.0% vs. 23.3%; *p* = 0.33). Furthermore, the rates of repeat surgery did not differ between patients in the pre-IVOS and the post-IVOS groups (11.0% vs. 8.7%: *p* = 0.62). Overall, the number of patients who required percutaneous drainage within 90 days of discharge (4/203, 1.9%), or who died within 90 days of discharge (9/203, 4.4%) were small, which precluded further comparisons. 

## 4. Discussion

Study findings demonstrated that pharmacists-led IVOS resulted in an increase in switch rate in the post-IVOS group when compared to the pre-IVOS group. Overall, no significant differences were observed in the clinical outcomes between patients who were switched by pharmacists, and patients who were not switched. Readmission rate and repeat surgery within 90 days of discharge, and the mean length of hospital stay were no different between the pre- and post-IVOS groups. This finding is in line with previous studies that assessed the clinical outcomes of IVOS of other antibiotics such as fluoroquinolones [20,30,31].

In the pre-IVOS group, 80/89 (89.9%) patients were eligible for a switch and did not receive a switch. This may be due to a common misconception that an early switch to oral treatment is not viable after intra-abdominal infections due to bowel dysmotility and impaired absorption [32]. However, early oral or enteral feeding enables early IV to oral switch, and is a useful therapeutic strategy for clinical improvement [32]. In contrast, only 28/93 (30.1%) of patients in the post-IVOS group who were eligible for a switch, did not receive a switch. Those patients were regarded as missed opportunities as they were not switched despite meeting the switch criteria. This could be explained by the initial lack of confidence of some pharmacists to initiate a switch at the beginning of the programme, and it was seen in the steady increase in switch rate as the study period progressed. Similar barriers have been seen elsewhere, where pharmacists voiced concern that a pharmacist-led IVOS switch might occur without proper patient clinical assessment, junior pharmacists would make therapeutic decisions beyond their clinical expertise, and the medical team would criticise their clinical judgements [5]. As in our study, concerns were addressed through discussions, in-service training, and ad hoc support by the study authors. This meant greater participation in the service and saw a steady increase in the switch rate (e.g., 16.7% to 77.8%).

The implementation of pharmacist-led IVOS of metronidazole did not lead to a reduction in the length of hospital stay. Due to the common practice of co-prescribing another IV antibiotic, such as cefuroxime, with metronidazole in intra-abdominal infections [33,34,35,36], patients receiving the combination remain in hospital until either the course is completed, or until the other IV antibiotic can be changed to another available/suitable oral antibiotic. The data for patients who received a combination of cefuroxime and metronidazole was not collected in our study. Additionally, there more complex, co-morbid and sicker patients needing longer hospital stay, and longer duration of IV metronidazole in the post-IVOS which have resulted in wider standard deviations compared to the Pre-IVOS group. Studies reported mixed results regarding length of hospital stay [14,19]; those that have shown a significantly reduced length of hospital stay have assessed antimicrobials that were used as a single agent therapy. However, a sub-analysis of the duration of IV metronidazole use in the post-IVOS group showed that patients who were switched by a pharmacist had shorter duration of IV metronidazole use when compared to those who were not switched by a pharmacist.

Surgical infections remain a major source of postoperative complications and hospital readmission contributing to between 30% and 40% of all readmissions in the United States [37,38]. In our study, the difference in the rates of readmission or repeat surgery within 90 days of discharge between patients who received a pharmacist-led switch and patients who did not have a switch by a pharmacist in the post-IVOS group was not statistically significant (*p* = 0.56). Other studies assessing clinical outcomes reported no statistical differences in clinical success between the intervention and the control groups [8,12]. Furthermore, early IVOS was as effective as continued IV treatment in terms of clinical outcomes in intra-abdominal infections [32,39,40], which is consistent with our findings. 

Our study had several limitations. As a single site, limited to General Surgery, Gynaecology, and Gastroenterology specialties, and with low cohort numbers, findings may not be extrapolated to a different site or population. Selection bias could have been introduced by pharmacists involved in initiating the switch and collecting data, evidenced by poor initial switch rate, which gradually increased with study duration and ad hoc support. The study did not take into consideration whether metronidazole was being used for the prophylaxis or treatment of an infection. Prophylactic metronidazole use is often for 24 to 48 h, whereas treatment courses are often longer, affecting the length of hospital stay, and could have resulted in discrepancies between pre- and post-IVOS data if more patients were prescribed prophylactic metronidazole in one group.

To our knowledge, this is the first published study in which pharmacist-led IVOS of metronidazole was evaluated in adult patients. Our results suggest that a pharmacist-led switch strategy does not affect clinical patient outcomes and demonstrates a successful introduction of a pharmacist-led switch protocol to deliver an AMS strategy. Despite the study limitations, we are confident that the pharmacist-led switch service had a positive impact in our hospital in increasing the metronidazole IVOS rate. This service was integrated into clinical pharmacists’ daily task as part of their clinical assessments of patients. The number of switches, and patient’s Medical Record Number is documented by clinical pharmacists using QualtricsXM. The percentage of switch rate (% metronidazole IVOS rate) has been included in the Pharmacy Department’s Key Performance Indicators and reported fortnightly to the clinical pharmacy staff during the Clinical Pharmacy Services meeting. Training for new staff on the IVOS programme was provided during their Infectious Diseases Orientation by the Infectious Diseases Pharmacist. 

The results of this study may help to instil more confidence in clinicians and pharmacists to continue to implement this service and consider expanding the service to include more antimicrobials such as penicillins and fluoroquinolones. Further studies of pharmacist-led IVOS implementation is recommended in other settings and sites. Future research could also investigate other clinical outcomes such as complications associated with IV-line administration.

In summary, our data have demonstrated successful implementation of the hospital-approved pharmacist-led IVOS service with a steady increase in the switch rate and no impact on the clinical outcomes.

## Figures and Tables

**Table 1 antibiotics-11-01303-t001:** Patient demographics and clinical outcomes.

Characteristics	Pre-IVOS (N = 100) n	Post-IVOS (N = 103) n
Median age (years), (range)	58 (21–88)	60 (18–94)
**Sex**		
Male	60	54
Female	40	49
**Ethnicity**		
Māori	13	10
Pacific Peoples	7	10
New Zealand European	48	48
Other	32	35
Mean Length of stay ± SD (days)	12 ± 17.2	13 ± 17.9
Median Length of stay (days)	5	7
Mean duration of IV metronidazole ± SD (days)	3.6 ± 2.8	3.4 ± 4.4
Number of IV metronidazole doses	8.7 ± 6.8	7.8 ± 8.5
**Site of infection**		
Colorectal	53	41
Hepatobiliary	28	38
Small bowel	3	3
Other	16	22
**Repeat surgery within 90 days of discharge**		
Pharmacist-led IVOS	0	5
No switch	11	4
**Readmission within 90 days of discharge**		
Pharmacist-led IVOS	0	15
No switch	18	9
**Percutaneous drainage procedures within 90 days of discharge**	1	3
**90-day all-cause mortality**	5	4
**Treatment subgroup**		
IV only (eligible for a switch)	82 (80)	30 (28)
IVOS	7	61
IVOS then reverted to IV	2	4
Oral only	9	8

## Data Availability

There is no public access to all data generated or analysed during this study to preserve the privacy of patients. The dataset that supports the conclusions is available upon request to the corresponding author.

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
