# Peer review of "The Impact of a Pharmacist-Led Intravenous to Oral Switch of Metronidazole: A Before-and-After Study"

_antibiotics, 2022, doi:10.3390/antibiotics11101303_

Round 1
Reviewer 1 Report
First and foremost, I would like to commend the authors for conducting this study that addresses an important issue that could impact patient outcomes. The following comments are worth attention before considering this manuscript for publication:
· Lines 131 - 137: The study included 103 patients in the post-IVOS group. However, the authors reported the switch of metronidazole by pharmacists out of 93 patients (i.e., 63/93 (68%)) and did the same thing when they reported the switch by physicians (i.e., 2/93 133 (2%)). No explanation was provided for the missing ten patients in these numbers. Lowering the denominator makes the percentage appear falsely high. I advise using a consistent denominator or providing an explanation when not doing so.
· Lines 138 - 140 and 190 - 192: The authors reported that IVOS led to a shorter duration of IV metronidazole use (3.64 ± 2.88 days vs. 3.43 ± 4.46 days, p=0.34; 8.75 ± 6.83 doses vs. 7.82 ± 8.50 doses, p=0.19). However, the standard deviation for the post-IVOS is wider than the pre-IVOS, making the upper end of the post-IVOS higher than the pre-IVOS (i.e., 7.89 vs. 6.52 days and 16.32 vs. 15.58 doses, respectively). The authors did not explain these findings in the discussion section.
Author Response
Reviewer 1
First comment
(Lines 131 - 137: The study included 103 patients in the post-IVOS group. However, the authors reported the switch of metronidazole by pharmacists out of 93 patients (i.e., 63/93 (68%)) and did the same thing when they reported the switch by physicians (i.e., 2/93 133 (2%)). No explanation was provided for the missing ten patients in these numbers. Lowering the denominator makes the percentage appear falsely high. I advise using a consistent denominator or providing an explanation when not doing so).
Reply
Thank you for your comment. The number of patients in the post-IVOS period was 103. However, 8 patients were prescribed oral metronidazole from the beginning of the antibiotic course. i.e., they didn’t receive IV metronidazole and didn’t require a switch, and 2 patients were not eligible for a switch. hence, our analysis only included those patients that were eligible for a switch i.e., 93 patients.
The following sentence was added to lines (139-140) provide clarity for the reader: “In the post-IVOS period, 93/103 (90.3%) were eligible for IVOS; of theses, pharmacists switched 63/93 (67.7%), and doctors switched 2/93 (2.2%)”.
Second comment
Lines 138 - 140 and 190 - 192: The authors reported that IVOS led to a shorter duration of IV metronidazole use (3.64 ± 2.88 days vs. 3.43 ± 4.46 days, p=0.34; 8.75 ± 6.83 doses vs. 7.82 ± 8.50 doses, p=0.19). However, the standard deviation for the post-IVOS is wider than the pre-IVOS, making the upper end of the post-IVOS higher than the pre-IVOS (i.e., 7.89 vs. 6.52 days and 16.32 vs. 15.58 doses, respectively). The authors did not explain these findings in the discussion section.
Reply
Thank you for your comment. Wider standard deviation can be explained by having more complex, co-morbid and sicker patients needing longer hospital stay and longer duration of IV metronidazole in the post-IVOS.
The following sentence was added to Lines 195-197 to explain the variation in standard deviation “Also, there more complex, co-morbid and sicker patients needing longer hospital stay, and longer duration of IV metronidazole in the post-IVOS which have resulted in wider standard deviations compared to the Pre-IVOS group”.
Also,
The following sentence was added to lines (200-203) to provide clarity to the reader
“However, a sub-analysis of the duration of IV metronidazole use in the post-IVOS group showed that patients who were switched by a pharmacist had shorter duration of IV metronidazole use when compared to those who were not switched by a pharmacist”.
Reviewer 2 Report
Overall, the work explores the important aspect of antimicrobial stewardship interventions to reduce costs and use of antimicrobials.
Importantly, intravenous to oral therapy switch is essential not only to reduce the hospital stay, but also to reduce complications associated with intravenous route.
The study is well presented and linear, however few points should be revised.
1) The specific choice of targeting Metronidazole for this intervention should be explained.
2) Personnel costs (number of pharmacists involved and working hours requested) should be included
3) Did Authors recorded complications associated with intravenous route of administration? This could be an interesting outcome to explore.
4) Please elucidate further projects in your Institution: what did Authors think should be the next step?
5) Please discuss the sustainability over time of these kind of projects: in many setting this point hamper the implementation of Antimicrobial Stewardship programs.
Author Response
Reviewer 2
Comment 1
The specific choice of targeting Metronidazole for this intervention should be explained.
Reply
Thank you for your comment. The following sentence was added to lines 69-71 to explain the reason for choosing metronidazole. “Metronidazole was chosen in this study due to its high bioavailability (>90%) and tissue penetration to infection sites [23], and the ability to tolerate oral food or medicines is the sole eligibility criteria for switching IV metronidazole to oral for non-septic patients”.
Comment 2
Personnel costs (number of pharmacists involved and working hours requested) should be included
Reply
We have now included the number of pharmacists in line 99. As pharmacists were undertaking this work as part of their routine clinical activities, we did not measure the personnel cost. We did consider the labour for the nursing staff, however, which is to be reported elsewhere.
Comment 3
Did Authors recorded complications associated with intravenous route of administration? This could be an interesting outcome to explore.
Reply
Thank you for your comment. In our study, we focussed on other clinical outcomes i.e. readmission readmission and/or repeat surgery with 90 days of discharge, and the length of hospital stay. Complications associated with intravenous administration is an outcome that can be considered for future research. The following sentence was added to lines 240-242:
“Future research could also investigate other clinical outcomes such as complications associated with IV-line administration.”.
Comment 4
Please elucidate further projects in your Institution: what did Authors think should be the next step?
Reply
Thank you for your comment. Our next step is to expand the IVOS to include more antibiotics such as to be automatically switched by pharmacists once pharmacists, medical staff, and nursing staff become familiar with the process. However, this will require extensive training for clinical pharmacists as other antibiotics such as penicillins and quinolones will require more clinical judgements. The following sentence was added to lines 237-239.
“The results of this study may help to instill more confidence in clinicians and pharmacists to continue to implement this service, and consider expanding the service to include more antimicrobials such as penicillins and fluoroquinolones”.
Comment 5
Please discuss the sustainability over time of these kind of projects: in many settings this point hamper the implementation of Antimicrobial Stewardship programs.
Reply
Thank you for your comment. This service was integrated into clinical pharmacists’ daily task as part of their clinical assessments of patients. The number of switches, and patient’s Medical Record Number is documented by clinical pharmacists using QualtricsXM. The percentage of switch rate (% metronidazole IVOS rate) has been included in the Pharmacy Department’s Key Performance Indicators and reported fortnightly to the clinical pharmacy staff during the Clinical Pharmacy Services meeting. Training for new staff on the IVOS programme was provided during their Infectious Diseases Orientation by the Infectious Diseases Pharmacist. This sentence was added to line 229-236.
Reviewer 3 Report
1) Methodology section is quit complex so suggestion is to modify.
2) How many patients in study either 200 or 203? two reading found?
3) Its good to discuss the switch criteria in methodology?
4) Results must be discussed in tables rather than in text.
5) The results are not completed.
6) Discuss new must be highlighted in paper.
Author Response
Reviewer 3
Comment 1
Methodology section is quit complex so suggestion is to modify.
Reply
Thank you for the suggestion. It is not clear which part of the methods the reviewer is referring to; however, authors have reviewed the methods, including subheadings to improve the organization and understanding of this section.
Comment 2
How many patients in study either 200 or 203? two reading found?
Reply
An a priori inclusion target of 200 eligible patients (100 patients in each group) was used based on previous reports, however 203 patients were recruited (100 in pre-IVOS and 103 in post-IVOS).
Comment 3
It is good to discuss the switch criteria in methodology
Reply
Thank you for your suggestion. The switch criteria were mentioned between in lines 84-87.
“The metronidazole IVOS programme was implemented by the AMS committee using the following simple criteria: tolerating oral medicines or food in non-ICU wards, and requiring pharmacists to document the prescription change in the patient’s clinical record and inform the patient and responsible medical team.”
Comment 4
Results must be discussed in tables rather than in text
Reply
Thank you for your suggestion. The majority of the results are presented in Table 1, with further clarification in text. The only results that have not been included in the table are the duration and number of IV doses. These results have now been added to Table 1.
Comment 5
The results are not completed
Reply
Authors believe that all the results have been presented in this section. Could the reviewer please provide further clarification so that the comment can be addressed.
Comment 6
Discuss new must be highlighted in paper
Reply
Thank you for your comment. Our next step is to expand the IVOS to include more antibiotics such as to be automatically switched by pharmacists once pharmacists, medical staff, and nursing staff become familiar with the process. However, this will require extensive training for clinical pharmacists as other antibiotics such as penicillins and quinolones will require more clinical judgements. The following sentence was added to lines 237-239.
“The results of this study may help to instill more confidence in clinicians and pharmacists to continue to implement this service, and consider expanding the service to include more antimicrobials such as penicillins and fluoroquinolones”.
Round 2
Reviewer 3 Report
Paper looks good now.